# Towards Auditory Profile-Based Hearing-Aid Fittings: BEAR Rationale and Clinical Implementation

Raul Sanchez-Lopez [1,2,3,4,*], Mengfan Wu [2,4,5,6], Michal Fereczkowski [5,6], Sébastien Santurette [7], Monika Baumann [7], Borys Kowalewski [8], Tobias Piechowiak [9], Nikolai Bisgaard [9], Gert Ravn [10], Sreeram Kaithali Narayanan [11], Torsten Dau [1] and Tobias Neher [5,6,*]

1 Hearing Systems Section, Department of Health Technology, Technical University of Denmark, 2800 Kongens Lyngby, Denmark
2 Hearing Sciences, Mental Health and Clinical Neurosciences, School of Medicine, University of Nottingham, Nottingham NG7 2RD, UK
3 Interacoustics Research Unit, 2800 Kongens Lyngby, Denmark
4 National Institute for Health and Care Research (NIHR), Nottingham Biomedical Research Centre, Nottingham NG7 2UH, UK
5 Institute of Clinical Research, Faculty of Health Sciences, University of Southern Denmark, 5230 Odense, Denmark
6 Research Unit for ORL—Head & Neck Surgery and Audiology, Odense University Hospital & University of Southern Denmark, 5230 Odense, Denmark
7 Centre for Applied Audiology Research, Oticon A/S, 2765 Smørum, Denmark
8 WS Audiology A/S, 3540 Lynge, Denmark
9 GN Hearing A/S, 2750 Ballerup, Denmark
10 Force Technology A/S, 2605 Aarhus, Denmark
11 Department of Electronic Systems, Aalborg University, 9220 Aalborg, Denmark
* Correspondence: rsalo@dtu.dk (R.S.-L.); tneher@health.sdu.dk (T.N.)

**Abstract:** (1) Background: To improve hearing-aid rehabilitation, the Danish 'Better hEAring Rehabilitation' (BEAR) project recently developed methods for individual hearing loss characterization and hearing-aid fitting. Four auditory profiles differing in terms of audiometric hearing loss and supra-threshold hearing abilities were identified. To enable auditory profile-based hearing-aid treatment, a fitting rationale leveraging differences in gain prescription and signal-to-noise (SNR) improvement was developed. This report describes the translation of this rationale to clinical devices supplied by three industrial partners. (2) Methods: Regarding the SNR improvement, advanced feature settings were proposed and verified based on free-field measurements made with an acoustic mannikin fitted with the different hearing aids. Regarding the gain prescription, a clinically feasible fitting tool and procedure based on real-ear gain adjustments were developed. (3) Results: Analyses of the collected real-ear gain and SNR improvement data confirmed the feasibility of the clinical implementation. Differences between the auditory profile-based fitting strategy and a current 'best practice' procedure based on the NAL-NL2 fitting rule were verified and are discussed in terms of limitations and future perspectives. (4) Conclusion: Based on a joint effort from academic and industrial partners, the BEAR fitting rationale was transferred to commercially available hearing aids.

**Keywords:** audiology; hearing rehabilitation; hearing aid

## 1. Introduction

Clinical hearing rehabilitation involves the sensory management of a hearing loss, which is typically addressed by means of hearing-aid (HA) fitting based on a set of audiometric thresholds. However, it is well known that there are hearing deficits that are only partially captured by an audiogram [1,2]. As such, conventional amplification cannot be expected to provide effective hearing loss compensation for speech understanding [3,4].

To address this shortcoming, the Danish "Better hEAring Rehabilitation" (BEAR) project recently developed strategies for individual hearing loss characterization and compensation. The characterization of hearing deficits is based on the concept of auditory profiling. Using various diagnostic tests, patients are stratified into four distinct groups called profiles A, B, C and D. This is achieved using a data-driven method [5]. This method was developed based on a relatively large dataset stemming from a sample of listeners with a wide range of hearing abilities who were tested with a comprehensive auditory test battery [6]. The test battery was afterwards reduced, based on considerations of cost-effectiveness and reliability, to arrive at the most informative diagnostic measures. These include loudness perception, speech intelligibility in noise, binaural hearing abilities, and spectro-temporal modulation sensitivity [7]. Furthermore, a profile-based HA fitting strategy called the BEAR strategy was proposed and evaluated in a pilot study [8], as summarized in Figure 1.

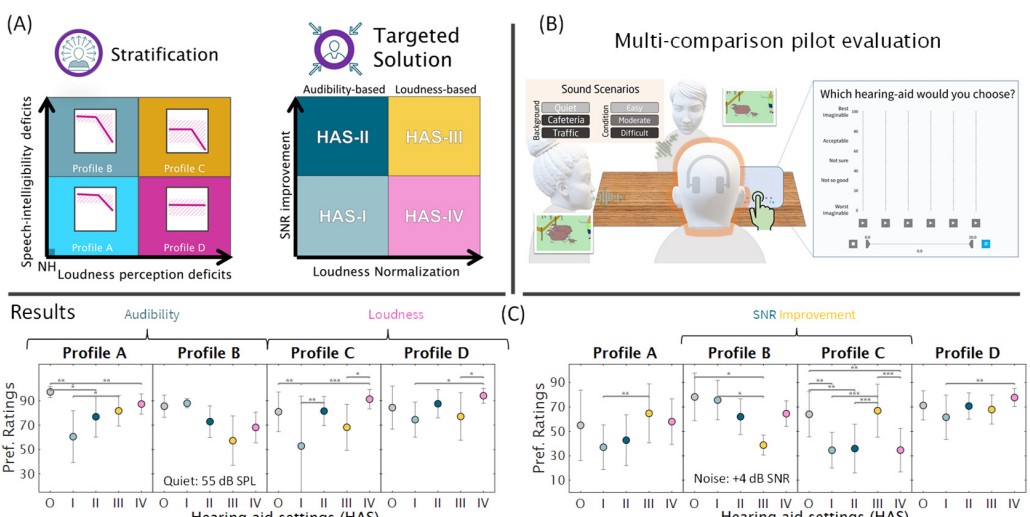

**Figure 1.** Overview of the pilot evaluation of the BEAR fitting strategy. (**A**) Stratification of the listeners into profiles A–D is based on two largely independent auditory dimensions, that is, speech intelligibility deficits and loudness perception deficits. Four tailored solutions (HAS I-IV) were proposed to compensate for these deficits using SNR improvement and loudness normalization, respectively. (**B**) Perceptual evaluation carried out with a HA simulator. (**C**) Results suggest that profile-C and -D listeners prefer their tailored solutions over a standard solution (HAS O), whereas profile-A and -B listeners do not show a clear preference. The image is based on the graphical abstract of [8].

Based on the results from this pilot evaluation, a large-scale randomized controlled trial was designed and carried out at two public hearing clinics. All participants underwent an initial evaluation, based on which they were stratified into one of the four auditory profiles. They were then randomly assigned to either the "BEAR" fitting strategy or a "current" fitting strategy. The participants assigned to the "current" strategy were fitted with HAs in accordance with current best clinical practice. That is, insertion gains (IGs) according to the "National Acoustic Laboratories—Non-Linear version 2" (NAL-NL2) fitting formula, which aims to maximize speech intelligibility and which was optimized based on empirical adjustments [9,10], were prescribed and verified using real-ear measurements (REM) [11]. Besides, earpieces and advanced feature settings were selected based on the recommendations made in the fitting software. In contrast, the participants assigned to the "BEAR" strategy were fitted with HAs depending on their profile membership. That is, IGs were prescribed based on the BEAR rationale [8] and verified using REM, and earpieces and advanced feature settings were chosen based on SNR improvement targets prescribed by the BEAR rationale [8].

The current report focuses on the clinical implementation of the BEAR fitting strategy. Its purpose is to demonstrate the transfer of this strategy to the hearing aids used in the large-scale clinical study. Technical measurements characterizing the HA fittings were performed in the laboratory to ensure that the targeted SNR improvement was achieved. Furthermore, REM performed in the clinics on the HA fittings made as part of the large-scale clinical study were evaluated.

## 2. The BEAR Rationale at a Glance

The BEAR rationale [8] includes the prescription of target gains and advanced feature settings, as summarized by the following formula:

$$\text{BEAR}(l, f, p) = 0.31 \cdot HTL(f) + \alpha(l, f, p) + \delta(p)$$

where $HTL(f)$ denotes the hearing thresholds at different frequencies, $\alpha(l, f, p)$ denotes gain correction factors applied at different input levels ($l$) and frequencies ($f$) and for the different profiles ($p$), and $\delta(p)$ denotes the SNR improvement to be applied for the different profiles ($p$). The constant factor 0.31 reflects the proportion of gain applied in relation to the HTL ("1-third rule"); for convenience, we use that specific value as it is specified in the original NAL formula [12,13]. The term $+\delta(p)$ does not represent an arithmetic sum but the SNR improvement. It does not affect the insertion gain.

To implement the BEAR rationale in three different commercially available devices, the key properties needing to be transferred first had to be identified. Table 1 summarizes the priorities chosen for the clinical implementation. To implement the BEAR rationale in three different commercially available HA devices, the key properties needing to be transferred had to be identified. Table 1 summarizes the priorities chosen for the clinical implementation. The SNR improvement was achieved by optimizing the settings of the directionality and noise reduction algorithms. The advanced features used in this study are shown in Appendix B (Table A1). Adaptive algorithms were not considered, and only features that aim for SNR improvement were activated in the BEAR fittings.

**Table 1.** Summary of the key properties (acoustic coupling, gain prescription, and SNR improvement) for implementing the BEAR rationale in commercially available devices.

| | HA Setting | Acoustic Coupling | Gain Prescription | SNR Improvement |
|---|---|---|---|---|
| BEAR strategy | A | Standard or custom ear-tips (open fit) | Maximize speech audibility | Small |
| | B | Custom ear-molds with venting | Maximize speech audibility | Large |
| | C | Custom ear-molds (closed fit *) | Loudness normalization | Large |
| | D | Custom ear-molds (closed fit *) | Loudness normalization | Small |
| Current strategy | O | Same as for BEAR | NAL-NL2 | Manufacturer default settings |

* With small (0.6–0.8 mm) vents.

## 3. Challenges to the Clinical BEAR Implementation

A number of challenges related to the clinical implementation of the BEAR rationale were identified, as listed below:

- Commercial fitting tools were not suited for the implementation, as they could not readily accommodate all required HA settings.
- While REM could be used for verifying IG targets, no clinically feasible method for verifying SNR improvement targets is currently available.

- The BEAR strategy had to be sufficiently different from the current HA fitting strategy to warrant a formal investigation into its perceptual benefits.
- The HA solutions for the four auditory profiles had to be sufficiently different from each other to warrant a formal investigation into their perceptual benefits.

## 4. Methods

For the final implementation, the BEAR rationale was slightly modified compared to the original proposal [8]. First, the proposed compression ratios had to be adjusted to settings that were practically realizable in the fitting software of the manufacturers. Second, the gain prescription was revised, so the soft input level corresponded to 55 (instead of 50) dB SPL. This was done to reduce the influence of background noise on the corresponding REM data.

The methods described below focus on a fitting tool developed for the clinical study, the procedure used for making real-ear measurements as part of the clinical study, and SNR improvement measurements made on an acoustic mannikin in preparation for the clinical study.

### 4.1. Clinical Fitting Tool

A clinically feasible fitting tool was developed to allow the commercial HAs to be fitted in accordance with the BEAR rationale. First, a Microsoft Excel sheet for calculating the BEAR target gains was prepared, into which the audiologists entered the audiometric thresholds at 0.25, 0.5, 1, 2, 4, and 8 kHz together with the profile of a given patient. The Excel sheet then generated a figure that was carefully designed to resemble the visual display in the REM system used for verification purposes (Affinity 2.0, Interacoustics, Middelfart, Denmark). Using the open-source software 'OnTopReplica' [14], the calculated BEAR gains were superimposed onto the visual REM display. The fitting tool and procedure are illustrated in Figure 2.

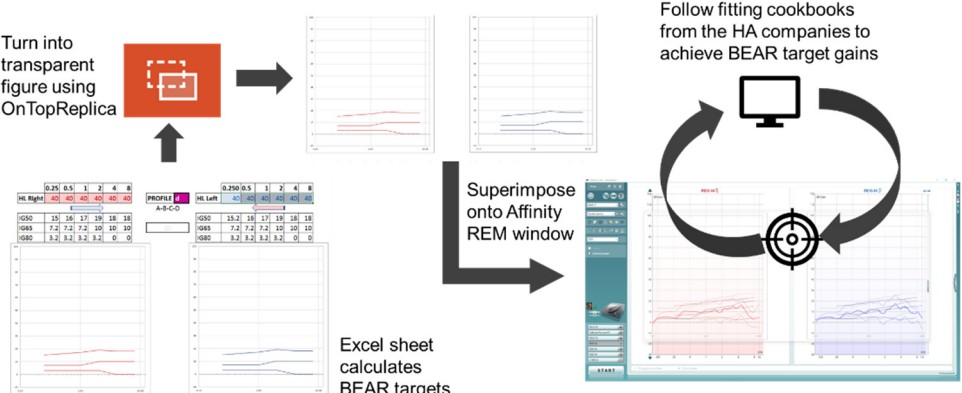

**Figure 2.** Illustration of the BEAR fitting tool and procedure developed for the clinical study.

The efficacy of the developed fitting tool and procedure was verified by performing several HA fittings on a CARL manikin (Ahead Simulations, Cambridge, ON, Canada). Care was taken to make the fitting process as straightforward as possible for the audiologists who handled the HA fittings in the clinical study. To accomplish this, detailed instructions were prepared to guide them through all necessary steps. Since three hearing aids from three different manufacturers were used, a set of instructions was needed for each device. Importantly, the instructions covered not only the gain adjustments, but also the activation and parameters of advanced features corresponding to the choices made for each profile in the BEAR fitting group, and the fitting protocol for the current fitting group.

### 4.2. Real-Ear Insertion Gain Measurements

As part of the clinical study, real-ear insertion gain (REIG) data were collected at 55, 65, and 80 dB SPL input levels to ensure close fits to the target. The International Speech Test Signal (ISTS, [15]) was used as the stimulus and played back from a loudspeaker approx. 1 m from the head of the participant. All recordings were carried out with the Interacoustics Affinity 2.0 system (Middlefart, Denmark). The REIG data were extracted as XML files and stored in an online database. Individual data files were processed to eliminate additional measurements performed during HA adjustment. Following the completion of the clinical study, information regarding the fitting strategy (current vs. BEAR) and auditory profile (A-D) was obtained and combined with the REIG data.

The participants for the clinical trial were recruited at two university hospitals (in Aalborg and Odense). Two-hundred-and-five adults with bilateral symmetric sensorineural hearing loss, Danish as their primary language, and no prior HA experience were included. They were 45–83 years old (mean and standard deviation: 68.3 $\pm$ 7.5 years), and 54% of them were male. Some participants dropped out of the study after the first visit. In total, 165 participants completed the study. At the first visit, the participants completed a clinical test battery for auditory profiling [7], based on which they were classified into a given profile. The distribution of the four auditory profiles was as follows: 53 profile-A, 92 profile-B, 14 profile-C, and 6 profile-D. There were 82 participants fitted according to the BEAR strategy and 83 participants fitted according to the current strategy. Within each profile, the distribution of the two fitting strategies was roughly equal.

### 4.3. SNR Improvement Measurements

To characterize the SNR improvement, electroacoustic measurements were performed in an IEC-standardized listening room at the Technical University of Denmark with a free-field setup with five loudspeakers placed in a circle with a radius of 1.5 m. At the center of the loudspeaker array, a head-and-torso simulator (HATS, type 4128, Brüel & Kjær, Nærum, Denmark) with pinnae (DZ9626-7, Brüel & Kjær, Nærum, Denmark) was placed (see Figure 3). The HATS was fitted with the test hearing devices and custom-made earpieces. The target speech signal (i.e., the ISTS) was played back from the frontal loudspeaker (0° azimuth) and so was an uncorrelated 4-talker babble noise from the other four loudspeakers (±45° and ±135° azimuth). The noise signals were calibrated to produce a sound pressure level of 70 dB SPL at the listening position.

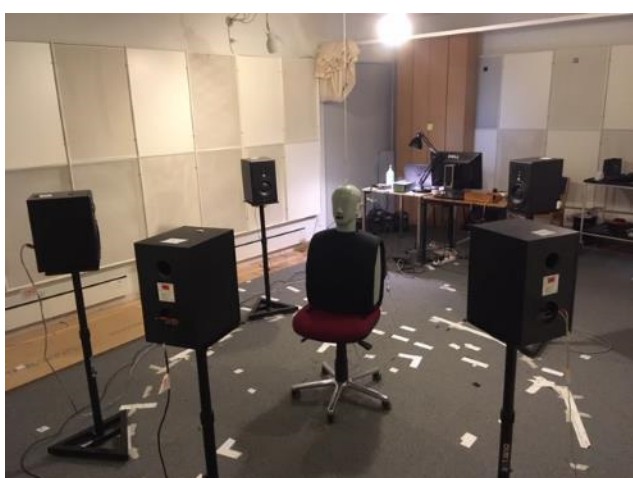

**Figure 3.** Picture of the test setup used for performing the SNR improvement measurements.

The measurements consisted of a series of 30 s recordings. The first 10 s of each recording were discarded. In this way, it was ensured that the advanced HA features had reached a steady state. To calculate the achieved SNR improvement, the Hagerman–Olofsson method [16] was used to separate the speech and noise signals on the HA output

side. The signals from the left and right microphones were both used in the analyses by concatenating them. First, the power spectral density of the target and noise signals was estimated in 18 one-third-octave bands with center frequencies according to (ANSI S3.5-1997). The average SNR was then calculated as the difference between the power spectral density of the target and noise signals averaged across all bands.

An additional reference recording was made with the unaided HATS. The SNR improvement, $\Delta SNR_{avg}$, was then calculated as the difference between the SNR estimated for each of the HA settings and the SNR from the unaided condition.

## 5. Results

### 5.1. REIG Measurements

Figure 4 shows mean REIGs measured at 55, 65, and 80 dB-SPL input levels. Each panel shows a comparison of the current fitting vs. one of the profile-based HA fittings (A, B, C, or D). While no differences between the two strategies are apparent for the profiles fitted according to considerations of audibility maximization (A and B), there are clear differences for the profiles fitted according to loudness normalization considerations (C and D). In both cases, the BEAR strategy provides less amplification for all input levels. This is especially clear at 80 dB SPL, where there is very little amplification below 4 kHz. Also, there are apparent gain differences between 65 and 80 dB-SPL input levels, corresponding to large compression ratios, as intended.

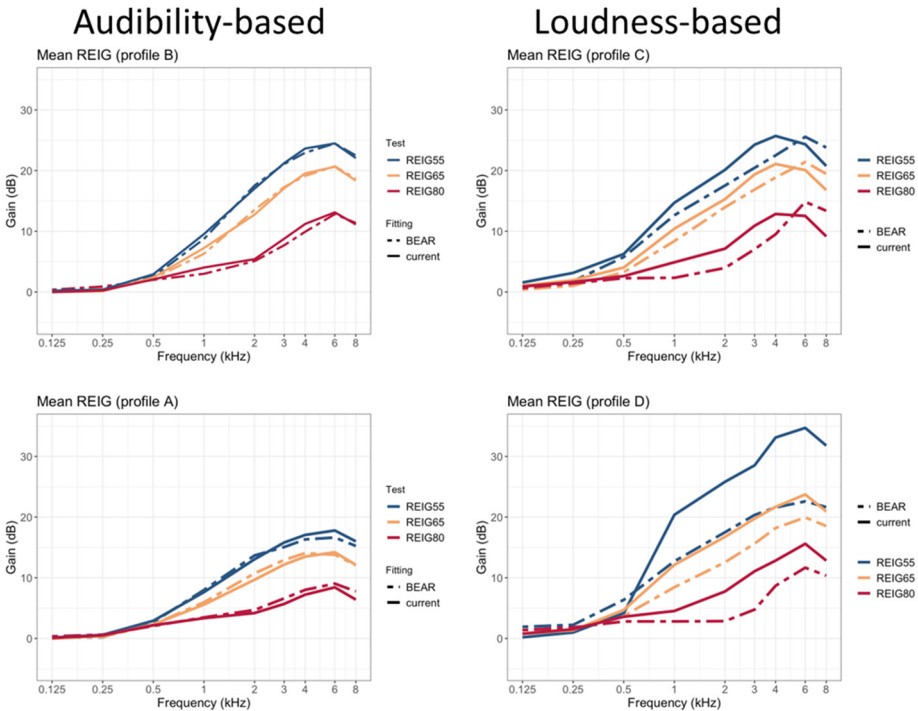

**Figure 4.** Mean real-ear insertion gains measured as part of the clinical study. Each panel shows data for one profile. Measurements made on participants fitted with the "current" strategy are shown with solid lines, while those made with the "BEAR" strategy are shown with dashed lines. The blue, yellow, and red lines represent input levels of 55, 65, and 80 dB SPL, respectively.

As there were only six participants in profile D, the differences in REIGs between the two fitting strategies can be largely explained based on differences in individual audiograms. To enable a comparison of the differences in gain between the two strategies for profile D, the gains measured for participants who received a BEAR fitting are compared with NAL-NL2 target gains for the same participants (i.e., for the same audiograms). This is illustrated in Appendix A. Overall, profile D is characterized by more gain at low frequencies and greater compression ratios in case of the BEAR strategy.

*5.2. SNR Improvement Measurements*

Figure 5 shows $\Delta\text{SNR}_{avg}$ values for three input SNRs ($-5$, 0, $+5$ dB) and the three HAs that were used in the clinical study (HA1, HA2, HA3). Each panel corresponds to one of the profiles. HA1 provided hardly any SNR improvement for profile A, but a notable SNR improvement (~3 dB) was seen for profile C. HA2 and HA3 provided 1–2 dB of SNR improvement for profiles A and D and >2 dB SNR improvement for profiles B and C. In terms of dependencies on the input SNR, there were only clear differences for profile B, especially with HA1, for which the SNR improvement decreased with increasing input SNR. This could have been a consequence of HA1 using fast-acting compression.

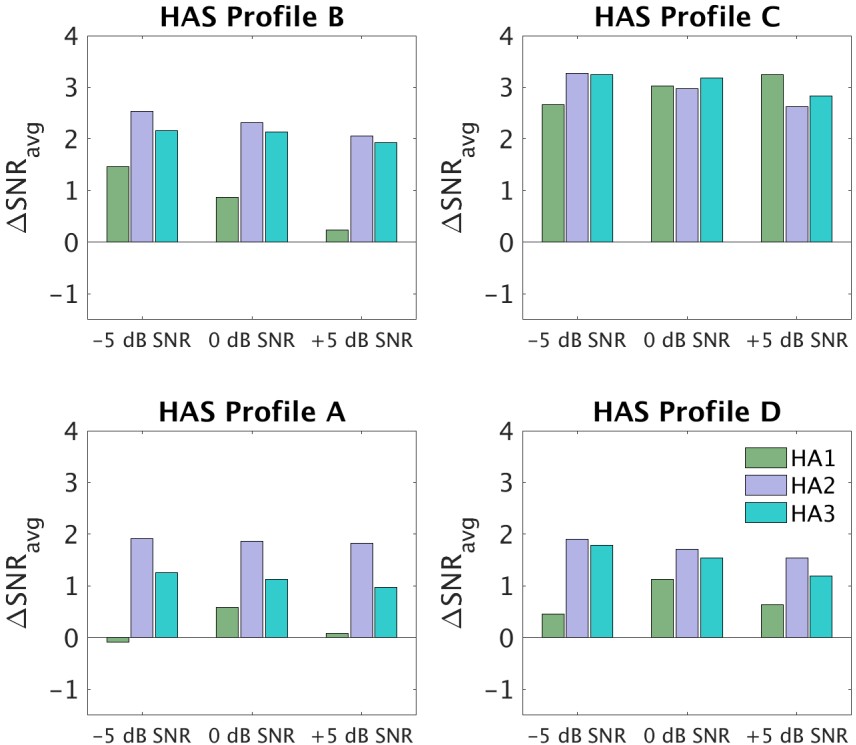

**Figure 5.** SNR improvement ($\Delta\text{SNR}_{avg}$) measured at three input SNRs ($-5$, 0, $+5$ dB) for the four auditory profile-based HA fittings (A–D) and three different HAs (HA1, HA2, HA3).

## 6. Discussion

*6.1. Differences between HA Settings for the Four BEAR Profiles*

The present study focused on challenges and solutions for implementing the BEAR fitting strategy in real HAs. The results of the technical evaluation showed that the three HAs were able to provide SNR improvement targets in accordance with the BEAR strategy. This was especially true for HA1, which did not provide any SNR improvement for profiles A and D but a substantial SNR improvement for profile C and, to a lesser extent, profile B at lower input SNRs. Overall, it was therefore possible to find advanced feature settings that fulfilled the requirements.

The REIG data were collected as part of the clinical study. As expected, the IGs for HAS-A and HAS-B were very similar to the ones prescribed by NAL-NL2. This was because the BEAR rationale prescribes IG based on the same principles as NAL-NL2 (i.e., maximization of speech audibility) for these two profiles. In contrast, the IGs corresponding to HAS-C and HAS-D deviated substantially from NAL-NL2. The IGs prescribed by the BEAR rationale for HAS-C and HAS-D are based on empirical comparisons between the gains required for loudness normalization, based on loudness perception measurements. The goal here is to provide sufficient amplification to normalize loudness at soft and conversational input levels and to reduce amplification for signal inputs above 75–80 dB SPL. This is partly motivated by the expected presence of 'rollover' in profile-C listeners,

which can affect speech intelligibility at above-conversational levels [17]. To achieve this compression behavior, large compression ratios are required, which are not easily achievable with commercial HAs. The reason for this limitation is that large compression ratios can compromise sound quality. Given the HAs used here, it was difficult to confirm whether the prescribed IG was normalizing the loudness as intended. A valid alternative would have been to fit the HAs while performing loudness perception tests, as suggested in [18,19], to individualize compression parameters. However, it is important to note that the profile-based HA fittings investigated here do not support the idea of using loudness normalization for all users, but only for those belonging to profiles C and D. Overall, it was possible to overcome many of the limitations in the test devices and to transfer the key properties of the BEAR rationale to them.

*6.2. Limitations*

A practical realization of the BEAR fitting strategy could be found based on a joint effort from the academic and industrial partners. The main limitation was that no modifications to the HA fitting software could be made. Instead, a procedure combining real-ear gain measurements with an open-source software was chosen for the gain adjustments. However, this procedure can be difficult for clinicians to perform, and thus errors can occur along the way.

While the REIG data were obtained from individual participants as part of the clinical study, the SNR improvement data could only be obtained in the laboratory. This makes a comparison of the current and BEAR fittings difficult. Currently, there is no systematic method for characterizing advanced HA signal processing in real ears. Although there are techniques that can successfully quantify signal modification [16,20], they require the use of head and torso simulators and a spatial loudspeaker configuration. Therefore, there is a need for clinically viable procedures that can be used to perform real-ear SNR measurements [21]. Ideally, it should be possible for such procedures to be routinely performed in the clinics using realistic scenarios and while the HA is operating as intended [22].

## 7. Conclusions

The BEAR fitting rationale was implemented for use in a large-scale clinical trial. The joint efforts by the industrial and academic partners resulted in a procedure for HA fitting that allowed the investigation of profile-based HA fittings with commercially available devices. As expected, the differences in IG between the BEAR and current fitting strategies were only apparent for profiles C and D, while the differences in SNR improvement were apparent for all profiles. The BEAR fitting rationale is the first fitting strategy that prescribes not only gain targets but also the adjustment of advanced HA features.

**Author Contributions:** Contributions designated as all reflect the contributions of all authors. Conceptualization; methodology; software; validation; formal analysis; and investigation, all; resources: T.D., T.N., S.S., B.K., N.B., G.R., T.P., M.B.; data curation, R.S.-L., M.W., S.K.N.; writing—original draft preparation, R.S.-L., M.W., T.N.; writing—review and editing, ALL; visualization, R.S.-L., M.W.; supervision, T.D., S.S., M.F., T.N.; project administration, T.N.; funding acquisition, T.N., T.D. All authors have read and agreed to the published version of the manuscript.

**Funding:** Collaboration and support by Innovationsfonden (Grand Solutions 5164-00011B), Oticon, GN Resound, WS Audiology, and other partners (University of Southern Denmark, Aalborg University, the Technical University of Denmark, Force Technology, and Aalborg, Odense and Copenhagen University Hospitals) is sincerely acknowledged.

**Institutional Review Board Statement:** Not applicable.

**Informed Consent Statement:** Informed consent was obtained from all subjects involved in the study. The study was notified to the Regional Committee on Health Research Ethics for Southern Denmark (case no. S-20162000-64).

**Data Availability Statement:** The data presented in this study and the BEAR protocol are available upon reasonable request from the corresponding author. The data are not publicly available due to patient data privacy. Excerpts of the data might be shared after proper anonymization.

**Acknowledgments:** We thank J. Zaar and S. Laugesen for their help with the SNR improvement setup and analysis. We also thank D. Hammershøi, G. Loquet, O. Cañete, R. Ordoñez, and other BEAR colleagues who provided valuable input on the BEAR procedure. We also thank colleagues from WSA, GN, and Oticon who contributed to the realization of the hearing-aid fitting guidelines.

**Conflicts of Interest:** The authors declare no conflict of interest.

## Appendix A  Comparison between BEAR and Current REIGs in Profile-D Participants

The profile-D group was relatively small. Therefore, a comparison between the REIG at 65 dB SPL and the prescribed NAL-NL2 targets is presented here. Figure A1 shows the REIGs for the three participants who were fitted according to the BEAR strategy, together with the calculated NAL-NL2 gains.

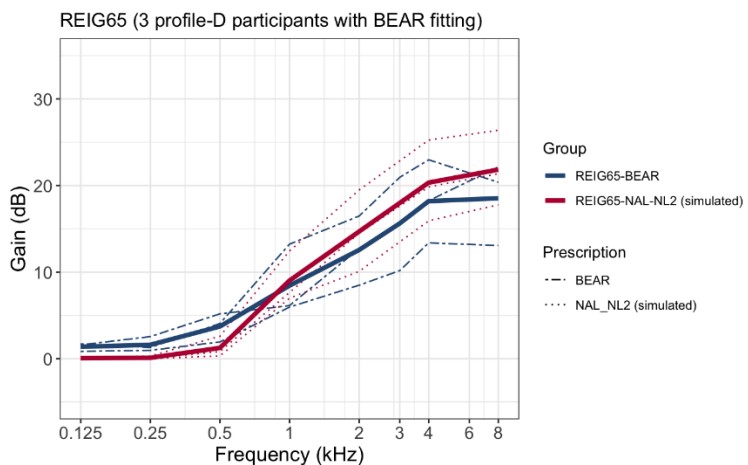

**Figure A1.** Mean REIGs at 65 dB SPL averaged across ears for three profile-D participants according to the BEAR strategy (blue lines). The red lines show calculated NAL-NL2 target gains.

## Appendix B  Hearing Devices and Advanced Features

**Table A1.** Settings of the advanced features for each profile-based hearing-aid fitting.

|  | A | B | C | D |
|---|---|---|---|---|
| Oticon Opn S1 | Dir: Open Automatic Low transition NR simpler = 0 dB NR complex = −5 dB | Dir: Open Automatic Medium transition NR simpler = −3 dB NR complex = −7 dB | Dir: Open Automatic Very High transition NR simpler = −3 dB NR complex = −9 dB | Dir: Open Automatic Low transition NR simpler = 0 dB NR complex = −5 dB |
| Widex Evoke 440 | Urban program. Speech and noise mode: Noise reduction comfort | Default Urban program settings | Default Impact program settings | Urban program. Speech and noise mode: Noise reduction comfort |
| GN Linx Quattro 9 | Omni NTII: Off | Fixed Dir. NTII: Strong | Fixed Dir. NTII: Strong | Omni NTII: Off |

Dir: Directionality, NR: Noise reduction; NTII: Noise Tracker II.

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
