# Peer review of "Towards Auditory Profile-Based Hearing-Aid Fittings: BEAR Rationale and Clinical Implementation"

_audiolres, doi:10.3390/audiolres12050055_

Round 1
Reviewer 1 Report
Dear Authors,
I think this is an exciting and meaningful research. It brought us a new practical clinical hearing fitting rationale based on auditory profile. I would recommend it if the below issues are handled well.
1. In the abstract, the results part should be more specific like the difference between NAL-NL2 and BEAR and the SNR improvements that BEAR could achieve
2. In line 75, is NAL-NL2 the best clinical practice? As I know, DSLv5 is also widely used in clinical hearing fitting. In the reference below, it is mentioned that DSLv5 performs better in medium frequencies.
Bertozzo, M.C. and Blasca, W.Q., 2019, August. Comparative analysis of the NAL-NL2 and DSL v5. 0a prescription procedures in the adaptation of hearing aids in the elderly. In _CoDAS_ (Vol. 31). Sociedade Brasileira de Fonoaudiologia.
3. In the results section, as the test is conducted with NAL-NL2 as the comparison, what is the advantage of using BEAR fitting against NAL-NL2? It seems only the differences between these two rationales are mentioned. Is there any feedback from subjects on which one they prefer?
4. In the BEAR rationale formula, it's α(l,f,p) // δ(p). But in the reference [8], it's α(l,f,p) + δ(p). Is it a typo?
Author Response
Reviewer / Author's Responses
I think this is an exciting and meaningful research. It brought us a new practical clinical hearing fitting rationale based on auditory profile. I would recommend it if the below issues are handled well.
Thank you for your positive feedback!
1. In the abstract, the results part should be more specific like the difference between NAL-NL2 and BEAR and the SNR improvements that BEAR could achieve.
To make this clear, we extended the text, specifying that the current "best practice" is based on NAL-NL2.
2. In line 75, is NAL-NL2 the best clinical practice? As I know, DSLv5 is also widely used in clinical hearing fitting. In the reference below, it is mentioned that DSLv5 performs better in medium frequencies.
Bertozzo, M.C. and Blasca, W.Q., 2019, August. Comparative analysis of the NAL-NL2 and DSL v5. 0a prescription procedures in the adaptation of hearing aids in the elderly. In _CoDAS_ (Vol. 31). Sociedade Brasileira de Fonoaudiologia.
Thank you for mentioning this. Although we decided to use NAL-NL2, the actual "best practice" was to systematically use a well-known fitting formula and verify it using real-ear measurements. To make this clear, we extended the text in the manuscript as follows:
"The participants assigned to the “current” strategy were fitted with HAs in accordance with current best clinical practice, that is, insertion gains (IGs) prescribed based on a well-known fitting formula, in this case the NAL-NL2 fitting rationale [9,10], and verified using real-ear measurements (REM)."
3. In the results section, as the test is conducted with NAL-NL2 as the comparison, what is the advantage of using BEAR fitting against NAL-NL2? It seems only the differences between these two rationales are mentioned. Is there any feedback from subjects on which one they prefer?
The participants completed several self-reported outcome measures and a test battery of aided performance as part of the study. However, those results are beyond the scope of the present paper and will be part of another publication.
4. In the BEAR rationale formula, it's α(l,f,p) // δ(p). But in the reference [8], it's α(l,f,p) + δ(p). Is it a typo?
Thank you for pointing this out: The term +δ(p) is not an arithmetic sum. The δ term represents the SNR improvement and it is not affecting the insertion gain. We used "+" in the revised version but also added an endnote explaining this.
Reviewer 2 Report
In their Manuscript "Towards auditory profile-based hearing-aid fittings: BEAR rationale and clinical implementation" Raul Sanchez-Lopez et al. describe the clinical implementation of the BEAR fitting strategy in real HAs as a large scale clinical trial with overall 165 participants.
Overall the manuscript is well written, the introduction section gives a clear overview of the current state of the art and the background, the methods section is clear, however needs some improvement regarding the included patient group, the results section again needs improvement including description of the included patients; the discussion section is precise, well based on the presented data and clear.
Despite some minor points the reviewer recommends acceptance of the manuscript after minor revision.
1) Descriptive data on the included patents are missing including age, gender, type and degree of hearing loss, additional diagnosis.
2) Figure 3: this figure is hard to read - resolution should be increased.
Author Response
Reviewer / Author's response
In their Manuscript "Towards auditory profile-based hearing-aid fittings: BEAR rationale and clinical implementation" Raul Sanchez-Lopez et al. describe the clinical implementation of the BEAR fitting strategy in real HAs as a large scale clinical trial with overall 165 participants.
Overall the manuscript is well written, the introduction section gives a clear overview of the current state of the art and the background, the methods section is clear, however needs some improvement regarding the included patient group, the results section again needs improvement including description of the included patients; the discussion section is precise, well based on the presented data and clear.
Thank you for your comments. We revised the manuscript accordingly.
Despite some minor points the reviewer recommends acceptance of the manuscript after minor revision.
1) Descriptive data on the included patents are missing including age, gender, type and degree of hearing loss, additional diagnosis.
We modified the text accordingly
"The participants were recruited at Aalborg and Odense University Hospitals. Two-hundred-and-five adults with bilateral symmetric sensorineural hearing loss, Danish as their primary language, and no prior experience with HAs were included in the study. Their age ranged 45-83 years (mean age: 68.3 ± 7.5 years) with a 54% male. Some participants dropped the study after the first sessions."
2) Figure 3: this figure is hard to read - resolution should be increased.
The purpose of Figure 3 is to illustrate how the use of the practical tool looks like. Since we cannot increase the resolution, and we do not have enough time to re-do the measurements and the figure again, we can only remove this figure if the editor considers that is best for the manuscript.
Reviewer 3 Report
Congratulations on completing a clinical trial of a solution for an important clinical question in our field. I have the following comments/questions and really look forward to reading your responses.
Please discuss what “advanced feature settings” means. Even if it is a combination of settings, listing them would help. Even though it is referenced in previous BEAR papers, it is referenced here many times and would greatly help understand what practical clinical knobs need to be turned to obtain these fits.
What other changes were made to the hearing aid programming in addition to altering the target gains in the BEAR fittings? It would be useful to list them in this version of the manuscript.
Would it be possible to show the spread of REIG values for the four profiles obtained in this study? If not in Figure 5, possibly in a supplementary table.
How would you propose to deal with the challenge listed on P3. L115-116? Since this involves multiple hearing aid settings while fitting, how does one realistically test this objectively?
Can you please comment on the challenges of implementing BEAR on 3 different hearing aids? Were the hearing aids used made by different manufacturers or were they different levels of product from the same manufacturer? It would help to provide slightly more detail without giving away the names/models, if that’s not acceptable to you.
Does Figure 6 present the average delta SNR values measured each type of device in each of the HA profiles? Can you please show the variability in this measure across listeners in each device/profile?
Please check spelling on inline citations for Interacoustics throughout the text.
P2. L 66-67. It is not clear from the figure how this statement can be made based on this figure. None of the scales in the perceptual evaluation portion are legible and there is no data point for Hearing Aid setting “O”. Even though this is possibly reproduced from another publication because it is presented here it should be readable.
P2. L81-82. What advanced feature settings were recommended based on the BEAR fitting strategy?
P3. L 94. Please check equation, should it be “+” the SNR improvement term?
Author Response
Reviewer / Author's Response
Congratulations on completing a clinical trial of a solution for an important clinical question in our field. I have the following comments/questions and really look forward to reading your responses.
Thank you for your positive feedback!
Please discuss what “advanced feature settings” means. Even if it is a combination of settings, listing them would help. Even though it is referenced in previous BEAR papers, it is referenced here many times and would greatly help understand what practical clinical knobs need to be turned to obtain these fits.
We added a table in Appendix B: Hearing devices and advanced features. In addition, we extended the main text as follows:
“To implement the BEAR rationale in three different commercially available HA devices, the key properties needing to be transferred had to be identified. Table 1 summarizes the priorities chosen for the clinical implementation. The SNR improvement was achieved by optimizing the settings of the directionality and noise reduction algorithms. The advanced features used in this study are shown in Appendix B (Table 2). Adaptive algorithms were not considered, and only features that aim for SNR improvement were activated in the BEAR fittings.”
What other changes were made to the hearing aid programming in addition to altering the target gains in the BEAR fittings? It would be useful to list them in this version of the manuscript.
The hearing-aid audiologists followed a protocol prepared by the three manufacturers, a set of instructions, to ensure that all hearing aids were programmed in the same way. To make this clear, we extended the text in the manuscript as follows:
“Care was taken to make the fitting process as straightforward as possible for the audiologists who handled the HA fittings in the clinical study. To accomplish this, detailed instructions were prepared to guide them through all necessary steps. These involved not only the gain adjustments, but also the activation of advanced features corresponding to the choices made for each profile in the BEAR fitting group.”
Would it be possible to show the spread of REIG values for the four profiles obtained in this study? If not in Figure 5, possibly in a supplementary table.
We considered this while revising the manuscript but had to discard the idea for reasons of time constraints (we were only given a few days to complete the revision). In any case, it would only be meaningful to show the spread for profiles A and B since the number of subjects in C and D was very small.
How would you propose to deal with the challenge listed on P3. L115-116? Since this involves multiple hearing aid settings while fitting, how does one realistically test this objectively?
The challenge referred to here is that "The BEAR strategy had to be sufficiently different from the current HA fitting strategy to warrant a formal investigation into its perceptual benefits."
The approach we followed was to control the BEAR HA fittings, that is, we manipulated or activated only those settings or algorithms that were in line with our hypotheses. This means that the BEAR fittings had deactivated many algorithms that by default were activated in the current fittings.
We added the following sentence to the manuscript: "Adaptive algorithms were not considered and only features that aim for SNR improvement were activated in the BEAR fittings."
Can you please comment on the challenges of implementing BEAR on 3 different hearing aids? Were the hearing aids used made by different manufacturers or were they different levels of product from the same manufacturer? It would help to provide slightly more detail without giving away the names/models, if that’s not acceptable to you.
The hearing aids were from three different manufacturers: Oticon, Widex and GN. We modified the manuscript accordingly:
“To accomplish this, detailed instructions were prepared to guide them through all necessary steps. Since three hearing aids from three different manufacturers were used, a set of instructions was needed for each device. Importantly, the instructions covered not only the gain adjustments, but also the activation of advanced features corresponding to the choices made for each profile in the BEAR fitting group and the protocol for the current fitting group.”
Does Figure 6 present the average delta SNR values measured each type of device in each of the HA profiles? Can you please show the variability in this measure across listeners in each device/profile?
The SNR improvement was measured in the laboratory on an acoustic mannikin and not different listeners. The three devices (HA1, HA2 and HA3) were tested with the parameters chosen for each profile (HAS-A, B, C and D) at three different input SNRs. There is no variability across measures because each bar corresponds to the average SNR improvement for each specific condition.
Please check spelling on inline citations for Interacoustics throughout the text.
Checked
P2. L 66-67. It is not clear from the figure how this statement can be made based on this figure. None of the scales in the perceptual evaluation portion are legible and there is no data point for Hearing Aid setting “O”. Even though this is possibly reproduced from another publication because it is presented here it should be readable.
Thank you for pointing out this. We modified the figure, so it is in line with the statement and the aspects discussed in the text are more legible.
P2. L81-82. What advanced feature settings were recommended based on the BEAR fitting strategy?
Please see Appendix B for details.
P3. L 94. Please check equation, should it be “+” the SNR improvement term?
Thank you for pointing this out: The term +δ(p) is not an arithmetic sum. The δ term represents the SNR improvement and it is not affecting the insertion gain. We used "+" in the revised version but also added an endnote explaining this.